# Investigating variability in microbial community composition in replicate environmental DNA samples down lake sediment cores

John K. Pearman[1]*, Georgia Thomson-Laing[1], Jamie D. Howarth[2], Marcus J. Vandergoes[3], Lucy Thompson[1], Andrew Rees[2], Susanna A. Wood[1]

**1** Coastal and Freshwater Group, Cawthron Institute, Nelson, New Zealand, **2** Victoria University of Wellington, Wellington, New Zealand, **3** GNS Science, Lower Hutt, New Zealand

* john.pearman@cawthron.org.nz

**Data Availability Statement:** The raw sequence reads were deposited in the NCBI short read archive under the accession number: PRJNA655562.

## Abstract

Lake sediments are natural archives that accumulate information on biological communities and their surrounding catchments. Paleolimnology has traditionally focussed on identifying fossilized organisms to reconstruct past environments. In the last decade, the application of molecular methodologies has increased in paleolimnological studies, but further research investigating factors such as sample heterogeneity and DNA degradation are required. In the present study we investigated bacterial community heterogeneity (16S rRNA metabarcoding) within depth slices (1-cm width). Sediment cores were collected from three lakes with differing sediment compositions. Samples were collected from a variety of depths which represent a period of time of approximately 1,200 years. Triplicate samples were collected from each depth slice and bacterial 16S rRNA metabarcoding was undertaken on each sample. Accumulation curves demonstrated that except for the deepest (oldest) slices, the combination of three replicate samples were insufficient to characterise the entire bacterial diversity. However, shared Amplicon Sequence Variants (ASVs) accounted for the majority of the reads in each depth slice (max. shared proportional read abundance 96%, 86%, 65% in the three lakes). Replicates within a depth slice generally clustered together in the Non-metric multidimensional scaling analysis. There was high community dissimilarity in older sediment in one of the cores, which was likely due to the laminae in the sediment core not being horizontal. Given that most paleolimnology studies explore broad scale shifts in community structure rather than seeking to identify rare species, this study demonstrates that a single sample is adequate to characterise shifts in dominant bacterial ASVs.

## Introduction

Understanding the lasting impacts of environmental decision making is fundamental to protecting and remediating Earth's biosphere. Long term insights are crucial to identifying and

**Funding:** This research was funded by the New Zealand Ministry of Business, Innovation and Employment research programme - Our lakes' health; past, present and future (C05X1707). The funders had no role in study design, data collection and analysis, decision to publish, or preparation of the manuscript. MJS works for GNS Science while JKP, GTL, LT and SAW work for Cawthron Institute. "The funders provided support in the form of salaries for the authors, but did not have any additional role in the study design, data collection and analysis, decision to publish, or preparation of the manuscript. The specific roles of these authors are articulated in the 'author contributions' section."

**Competing interests:** MJS works for GNS Science while JKP, GTL, LT and SAW work for Cawthron Institute. The funders provided support in the form of salaries for the authors, but did not have any additional role in the study design, data collection and analysis, decision to publish, or preparation of the manuscript. The specific roles of these authors are articulated in the 'author contributions' section. The commercial affiliations do not alter our adherence to PLOS ONE policies on sharing data and materials.

understanding ecological processes and inherent variability; however, biological monitoring rarely extends beyond a few decades [1]. Lake sediments accumulate information about autochthonous biological communities [2, 3] and their surrounding catchments [4, 5], enabling temporal insights into ecological processes [1]. Paleolimnology has traditionally focussed on identifying and enumerating organisms' remains, but in the last decade, advances in molecular methodologies have permitted the identification of soft-bodied organisms that typically do not leave "footprints" in the sediment record [6 and references within].

The genetic material present in environmental samples, such as sediment, can be defined as environmental DNA (eDNA). This DNA originates from a complex mixture of living cells, inactive cells, fragments of dead cells and extracellular DNA from a variety of sources including faeces, urine and saliva [7]. Techniques such as metabarcoding, which enable the identification of taxa from samples containing a mix of species using high throughput sequencing and DNA reference databases, have recently been applied to characterise bacteria, Archaea, microbial eukaryotes, plants and vertebrates in lake sediments [8–12]. The application of these techniques has allowed new insights into temporal trends over decades, centuries and even millennium. Their use has the potential to allow testing of ecological theories which require long term data sets [e.g. evolutionary dynamics, anthropogenic impacts 6, 13].

Before molecular techniques can be robustly applied to the analysis of biological communities in sediment cores, there are a number of methodological considerations that need to be addressed. For example, numerous studies using metabarcoding approaches have shown that DNA extraction efficiency and taxonomic coverage of microbial species in sediments is substantially different depending on the DNA extraction method applied [14–17]. A comprehensive review by Lear et al., [18] suggested that the most appropriate method for small sediment volumes was the DNeasy PowerSoil kit (Qiagen, Germany), which was used in the current study.

Beyond technical considerations, lake sediments have different compositions, which can affect the absorption of DNA; for example, humic acids and clay materials have a stronger binding capacity for DNA than other sediments [19]. Mineralogy, porewater pH, and temperature can also impact DNA retention and degradation in sediments [20, 21]. Reworking of sediments, and subsequent DNA transport, is known to occur through the movement of water, active growth of organisms and bioturbation [22]. However, initial research indicates that DNA leaching is minimal in saturated aquatic sediments where pore water movement is minimal [4, 23].

An area of methodological consideration that requires further investigation in lake sediment cores is heterogeneity in eDNA across the small spatial scales of depth slices of the cores. Spatial heterogeneity within surface sediment samples has been previously shown [24, 25] but while some work has indicated clusters of replicates down sediment cores [26], there has been limited investigation of how variability amongst replicates changes with depth. The investigation of variation within depth slices in sediment cores is especially vital as a large proportion of studies base their analysis on one or two DNA extractions from each depth slice [27–31].

The aim of the present study was to investigate bacterial community heterogeneity (16S rRNA metabarcoding) within depth slices. Sediment cores were collected from three lakes with differing sediment compositions. The deepest sediment in the cores was estimated to be up to 1,200 years old. Triplicate DNA extractions followed by bacterial 16S rRNA metabarcoding were undertaken on samples collected from core depth slices (1 cm width). We hypothesised that: (1) Replication would provide a better representation of the ASV richness within core depth slices; (2) a more complete inventory of microbial richness (accumulation curves plateau) would be obtained with increasing depth; (3) community structure and composition would be more similar within depth slices than between depth slices; and (4) that there would

be greater variability in the triplicates with increasing age of sediment due to differences in DNA degradation over time.

## Methods

### Sampling sites

The three lakes were chosen to cover a wide range of sediment types (S1 Fig). The map was produced in ggmap [32] using Map tiles by Stamen Design, under CC BY 3.0. Data by Open-StreetMap, under ODbL.

Lake Nganoke is a small (~3.1 ha), shallow (max. depth 2.3 m) lake located at 17 m above sea level (a.s.l) at the southern end of the Wairarapa Valley in the lower North Island of New Zealand (41˚21′20″ S, 175˚11′10″ E). The catchment of the lake (~174 ha) is incised into a terrace that has a maximum elevation of 100 m and is vegetated with high producing exotic grassland used for sheep and beef farming. The catchment lithology is characterised by early Pleistocene river gravels and late Pleistocene beach deposits [33]. Sediments accumulating on the lake floor are highly organic silts formed largely from autochthonous sedimentation.

Lake Paringa is a medium sized lake (~475 ha) situated at 16 m a.s.l with a maximum depth of 58 m that has formed in an over-deepened glacial trough (~17,000 years ago). It is located on the West Coast of the South Island of New Zealand (43˚43'10"S, 169˚24'8"E), within a large catchment (~6,000 ha) that drains the ~1420 m a.s.l high frontal range of the Southern Alps. The catchment is primarily vegetated in undisturbed temperate podocarp rainforest [34]. The lithology of the lake catchment is predominantly cataclasites, mylonites and schists of the Rakaia Terrane east of the Alpine Fault and Greenland group metasediments to the west of the fault [35]. Lake Paringa's sedimentary fill is characterized by a repeating sequence of three deposits comprising (i) co-seismic megaturbidites formed by shaking-induced subaqueous mass wasting; (ii) post-seismic hyperpycnite stacks formed during periods of elevated fluvial sediment flux from earthquake-induced landsliding; and (iii) inter-seismic layered silts formed between earthquakes when the catchment is relatively geomorphically quiescent [34].

Lake Pounui is a small (~46 ha), shallow (max. depth 9.6 m), lowland coastal lake situated at 10 m a.s.l about 30 km northeast of Wellington, New Zealand (41˚20'34"S, 175˚6'48"E). Lake Pounui is thought to have formed ~3,000 years ago by the damming of a stream valley by beach sediment and alluvial gravels [36]. Lake Pounui's catchment (627 ha) extends to an elevation of 470 m in the foot hills of the Rimutaka Ranges. The majority of hillslopes are vegetated by unmodified indigenous beech-podocarp forest (96%), with the remainder in pastoral land cover [37]. The catchment lithology is composed on late Pleistocene marine benches underlying beach deposits, loess and alluvial gravels south east of the Wharekauhau Thrust fault and Esk Head sand and mudstones north west of the fault [33, 36]; material eroded from these sources enters Lake Pounui [36, 38, 39]. The sediments accumulating on the lake floor are a mixture of allochthonous silts and very fine sands and autochthonous organic silts [36].

### Sample collection

In lakes Pounui and Nganoke sediment cores were retrieved using a Uwitech (Mondsee, Austria) gravity corer with 2-m polyvinyl chloride (PVC; 65-mm dia.) barrels. In Lake Paringa a 6-m sediment core was collected using a Mackereth corer (50-mm dia.; [40]). Barrels were cleaned with 2% bleach prior to coring. The collected sediment cores were stored at 4˚C in darkness for up to 4 weeks until sub-sampling for lakes Nganoke and Paringa. The samples from Lake Pounui were collected, subsampled and stored frozen (-80˚C) within 6 hours. A single core per lake was used for molecular analysis.

In the laboratory, the cores were sliced along the longitudinal plane, photographed, and described in detail including colour, texture and structure [41]. To prevent contamination, the top 2–3 mm of each half of the core was carefully removed with a sterile spatula. Slices of the cores were sampled at various depths from the top of the core (S1 Table and S2 Fig). Triplicate sub-samples (~ 0.5 g) were taken near the centre of the half-core using a sterile spatula and placed in separate tubes. Samples were kept frozen (-20°C) and in the dark until DNA extraction. Data on the age of sediment were sourced from other studies [34, 42, 43; S1 Table].

## DNA extraction, PCR and high throughput sequencing

Each step of the molecular analyses (i.e. DNA extraction, Polymerase Chain Reaction [PCR] set-up, template addition, PCR analysis) was conducted in dedicated separated sterile laboratories, with sequential workflow to ensure no cross-contamination. Rooms dedicated to DNA extraction, amplification set-up and template addition were equipped with ultra-violet (UV) sterilisation. UV sterilisation of the room and equipment was undertaken for 15 min before and after each use. The PCR set-up and template addition were always undertaken in laminar flow cabinets with HEPA filtration. Aerosol barrier tips were used throughout.

DNA was extracted from approximately 0.25 g of sediment using the DNeasy PowerSoil Kit (Qiagen, Germany) following the manufacturer's instructions on a QIAcube sample preparation robot (Qiagen). A negative extraction control, where no sediment was added to the extraction tube, was included for every 23 samples.

The V3-V4 region of the bacterial 16S rRNA gene was amplified by PCR using the bacterial specific primers 341F: 5'-CCT ACG GGN GGC WGC AG-3' and 805R: 5'-GAC TAC HVG GGT ATC TAA TCC-3' [44]. The primers included Illumina™ overhang adapters to allow dual indexing as described in Kozich et al., [45]. Triplicate PCR reactions were undertaken on each sample in 20-μL volumes. The reaction mixture consisted of 10 μL of $2 \times$ PCR MyFi™ Mix (Bioline), 1 μL of each primer (10 μM), 1.5 μL of bovine serum albumin (20 mg mL$^{-1}$) and 1.5 μL of template DNA. Cycling conditions were an initial 1 min 30 sec at 94°C before 30 cycles of 94°C for 30 s, 52°C for 30 s, 72°C for 45 s, with a final extension step at 72°C for 5 min. Negative PCR controls were run alongside the samples. The triplicate PCR replicates were pooled and 20 μL was cleaned and normalized using SequalPrep Normalization plates (ThermoFisher Scientific, USA), resulting in a concentration of ~1 ng mL$^{-1}$. The cleaned samples were sent to Auckland Genomics Facility for paired-end (2 x 250 base pairs (bp)) sequencing on an Illumina Miseq™ platform. Sequence libraries were prepared as detailed in the Illumina 16S metagenomics library prep manual (https://support.illumina.com/documents/documentation/chemistry_documentation/16s/16s-metagenomic-library-prep-guide-15044223-b.pdf). One deviation from the protocol was that, after the indexing PCR, 5 μL of each sample was pooled, and clean-up was undertaken on the pooled library instead of samples being individually cleaned. The concentration and quality of the library was quantified using a bioanalyzer. The library was diluted to 4 nM and denatured, and a 15% PhiX spike-in was added. The library was further diluted to a final loading concentration of 7 ρM; raw sequence reads were deposited in the NCBI short read archive under the accession number: PRJNA655562.

Primers were removed from the raw reads with cutadapt [46] allowing one mismatch. Sequences without primer sequences were discarded. Remaining sequences were processed with DADA2 [47] within the R framework [48]. Forward reads were truncated to 230 bp while reverse reads were truncated to 228 bp. The number of allowed maximum "expected errors" (maxEE) was two and four, to account for differences in sequence quality, for the forward and reverse sequences respectively. The first $10^8$ bp were used to construct a parametric error

matrix. Sequences were dereplicated and amplicon sequence variants (ASVs) were inferred based on this error matrix. After, inference singletons were discarded and remaining paired-end reads were merged with a maximum of 1 bp mismatch and a required minimum overlap of 10 bp. Chimeric sequences were removed from the analysis using the removeBimeraDenovo function within DADA2.

The resulting ASVs were taxonomically classified in DADA2 using the rdp classifier [49] against the SILVA 132 database [50] with a bootstrap of 70. The results were combined into a phyloseq object [51], and any ASVs classified as eukaryotes, chloroplasts or mitochondria were removed. Negative controls were assessed and read numbers for ASVs found in the negative controls were removed from the samples via subtraction [52]. After this quality check step on average greater than 99% of reads were retained in the lakes Nganoke and Pounui samples, and greater than 95% for Lake Paringa samples (S2 Table).

Accumulation curves were plotted showing the number of ASVs against sequencing depth for each replicate and for the replicates merged together using the R package ranacapa [53]. To compare samples within lakes, subsampling to an even sequencing depth was undertaken for each sample per lake (Lake Nganoke: 14,900 reads, Lake Paringa:18,800 reads, Lake Pounui: 9,800 reads). The number of observed ASVs was calculated per sample, using the phyloseq package [51] and plotted against depth to see changes in richness down the core. The number of shared ASVs and reads amongst the replicates was calculated to assess the similarity of the replicates within a slice. The community composition of each depth slice was assessed at the phyla level of taxonomic classification. Multivariate analysis, to see the clustering of replicates within slices, was undertaken on the samples using Bray Curtis dissimilarity matrices and visualized with non-metric multidimensional scaling (nMDS) plots. Each depth slice was assigned a number, increasing with depth, to assist with visualization of data. Smaller numbers represent the top of the core and larger numbers deeper slices in the core. Exact depths/ages are detailed in S1 Table. Comparisons of the mean similarity (1- Bray Curtis distance and 1- Jaccard distance) within and between depth slices were calculated using a Kruskal-Wallis test in R. The mean similarity (1- Bray Curtis distance) of the triplicates for each depth slice was calculated and plotted with a linear regression calculated to assess differences in similarity with depth. Figures were produced in R using the package ggplot2 [54].

## Results and discussion

The use of molecular techniques to characterise biological communities in environmental samples has been increasing over the past decade. There is an ongoing need to ensure an in-depth knowledge of the robustness and caveats of these techniques for specific applications. A total of 5,317,829, 1,880,055 and 1,827,617 sequences were obtained for sediment core samples from lakes Nganoke, Paringa and Pounui, respectively. After bioinformatically subsampling to an even depth, this resulted in 45,139 ASVs for Lake Nganoke, 19,540 ASVs for Lake Paringa, and 51,202 ASVs for Lake Pounui.

Metabarcoding samples often contain a high number of low abundance ASVs [55, 56], and large sequencing efforts are required to reveal the full level of diversity. This was observed in the present study and is highlighted by the accumulation curves which, in general, did not reach a plateau, especially in samples near the top of the core and for the merged replicates (Fig 1 and S3–S5 Figs). Similar results have been observed in other lake sediment studies that have used molecular methodologies [57, 58]. These pipelines have a finer scale of resolution than previous operational taxonomic unit methods [47]. It is however feasible that a portion of the diversity observed is due to PCR/sequencing errors that were not detected in the bioinformatics pipeline. The accumulation curves in the present study suggest that taking three

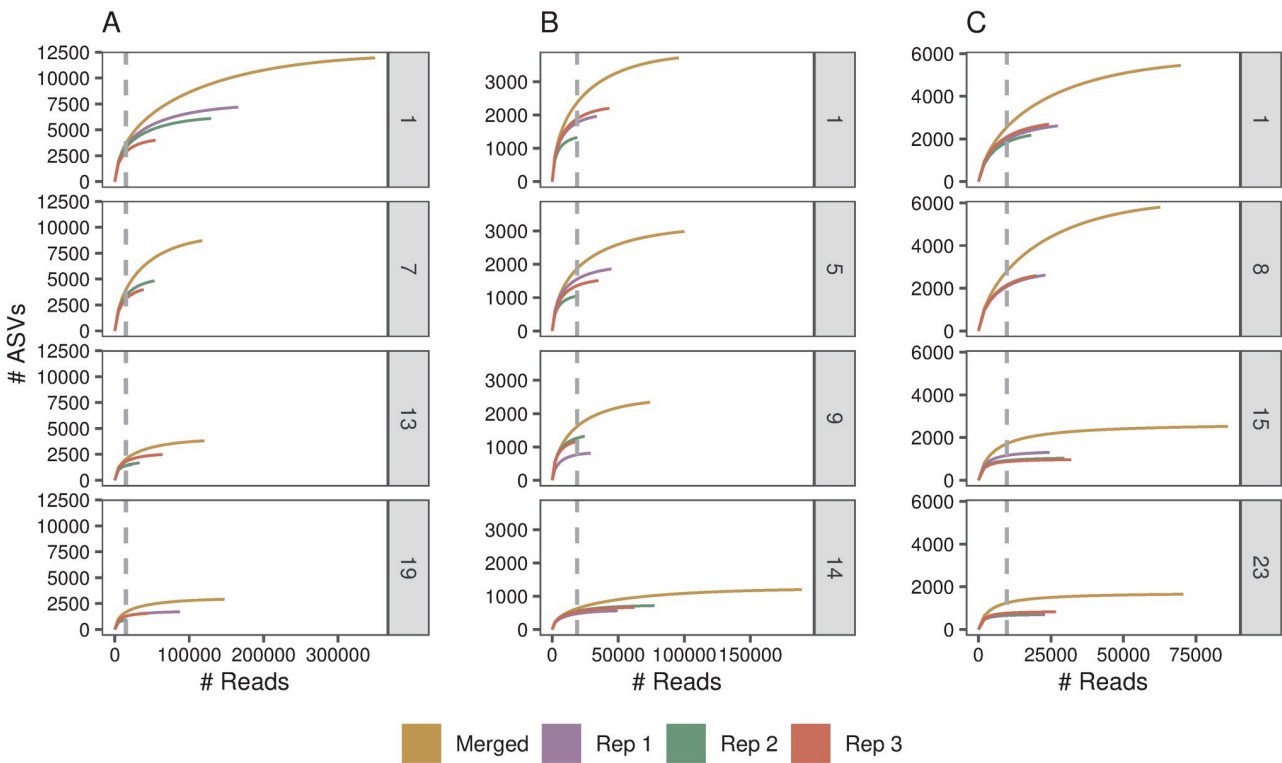

**Fig 1. Accumulation curves displaying the observed number of Amplicon Sequence Variants (ASVs) with increasing number of reads for four distinct depth slices (top of core, two slices at intermediate core depths and bottom of core (see S1 Table for depths of slices)) in sediment core samples taken from; A) Lake Nganoke, B) Lake Paringa, and C) Lake Pounui.** Accumulation curves are depicted for individual replicates and as a combination of replicates (merged). The level of sub sampling to an even depth used in the further analysis was indicated by the dashed vertical line. Plots for all slices are in S3–S5 Figs. Note: some lines overlap and therefore not all three are visible.

replicate samples is insufficient to capture the full extent of bacterial diversity in most sediment core slices, although a fuller inventory of the ASVs in deeper samples was achieved (Fig 1). Further investigation including more replication as well as the inclusion of technical replicates is required to improve the estimation of diversity within sediment core samples, especially in shallower samples.

The more complete inventory of ASVs in the deeper sections of the core could be due to the lower ASV richness observed with depth in the core (S6 Fig). This trend could be the result of a couple of factors. Firstly, at deeper depths there may be increased damage to the DNA present in the sediment through strand breakage, abasic sites, miscoding lesions and crosslinks [59, 60]. This can result in fewer amplifiable templates and, as the majority of the ASVs present in metabarcoding samples are rare and of low abundance, it is likely that a proportion of these would no longer be detected [6, 58]. Rates of DNA degradation are affected by a combination of abiotic (e.g. mineralogical composition, organic load, temperature and stable stratification) and biotic factors (e.g. degradation via DNase activity) and thus the effect of DNA degradation is likely to vary between lakes [6, 61]. Secondly, the samples taken in this study cover a time scale ranging from ~800 years (Pounui) to ~1200 years (Paringa) before present. Thus, it cannot be ruled out that alpha diversity is being affected by temporal changes, driven by pressures such as climate change and anthropogenic impacts, in the ecology of the lakes. Indeed, it has previously been shown that climate change has led to an increase in the diversity of photosynthetic microbial taxa [62, 63]. An in-depth analysis of the composition of the changes in

microbial communities is the subject of other research, but a high-level analysis shows temporal changes. For example, the phylum Bacteroidetes increases in relative abundance at the top of the core in all three lakes (S7 Fig). Other phyla, such as Cyanobacteria (Pounui) and Nitrospirae (Nganoke) also increase in relative abundance in recent times, indicating shifts in lake dynamics possibly related to increases in temperature [63] and nutrients [64], respectively. A combination of methods could be applied to obtain a better indication of sediment core richness, including increasing sequencing depth within a sample, undertaking multiple DNA extractions at a depth, or increasing the number of PCR replicates undertaken [30, 65].

The number of ASVs shared amongst the three replicates was lowest in Lake Pounui (6.5–12.2%), with Lake Paringa (11.7–25.3%) and Lake Nganoke (13.7–23.4%) having higher proportions of shared ASVs. There was no significant trend in the number of shared ASVs with depth (p > 0.05; S8A–S8C Fig). Although the number of ASVs shared amongst the three replicates within a depth slice was relatively low, the majority of reads were accounted for by shared ASVs. This indicates that within a depth slice the abundant community is being detected in all replicates, however there is a large number of rare and low abundance ASVs present as has been previously observed in environmental metabarcoding studies [55, 56]. In Lake Paringa, shared ASVs amongst replicates within a depth slice accounted for a maximum of 96% (range 76–96%) of the reads, compared to 69–86% in Lake Nganoke and 45–65% in Lake Pounui. Bioturbation, namely organism-induced mixing of sediment in both vertical and horizontal directions, might affect the numbers of shared reads within each lake. Lakes Nganoke and Pounui both have kākahi (freshwater mussel; *Echyridella menziesi*) which would likely result in higher bioturbation rates than in Lake Paringa. With lower levels of bioturbation, the stratigraphy would be less disturbed leading to more homogeneous replicates at any depth and consequently a higher shared number of reads. While lakes Nganoke and Pounui have higher levels of bioturbation, likely reducing the number of shared reads amongst triplicates, there is lower sedimentary layering in Lake Nganoke. Bioturbation is therefore likely to be mixing similar layers and distinct changes in the community would be reduced.

Similarity comparisons were undertaken to further explore differences within and between depth slices. The mean similarity of pairwise comparisons within depth slices (based on relative abundance data) was significantly higher than the mean of pairwise similarity comparisons between slices (1 –Bray Curtis distance; Nganoke: p < 0.001; Paringa: p < 0.001; Pounui: p < 0.001; Fig 2A–2C). A similar trend was noted when the presence/absence of ASVs was assessed (1 –Jaccard distance; Nganoke: p < 0.001; Paringa: p < 0.001; Pounui: p < 0.001; Fig 2D–2F). This indicates that the variability in bacteria communities within a depth slice is lower than amongst depths. Thus, this suggests that temporal trends can be discerned from microbial communities across small spatial scales in agreement with previous studies.

The non-metric multidimensional (nMDS) plots showed that depth slice replicates generally clustered together (Fig 3) although there was a temporal trend observed in all lakes. We anticipated that there would be marked differences in bacterial community composition and structure across depths in all lakes. Over the last ~1,000 years these lakes have been subjected to multiple natural and anthropogenic disturbances which would have impacted their microbial communities ([34, 43], Andrew Rees, Victoria University unpub. data). Studies on these taxonomic shifts and their potential drivers are the focus of ongoing studies. There were several notable exceptions to the clustering, for example, triplicates were relatively widely dispersed in Lake Paringa slices 9–11. Reasons for the lower homogeneity in these samples is unknown, and technical origins of this variation cannot be ruled out, but there could be greater sediment heterogeneity due to rapid changes in the landscape around the lakes (caused by impacts of human settlement or natural disturbances, i.e., earthquakes). In Lake Pounui slices 11, 19 and 20 (Fig 3) had a wider spread. Slices 19 and 20 correspond to a region of the

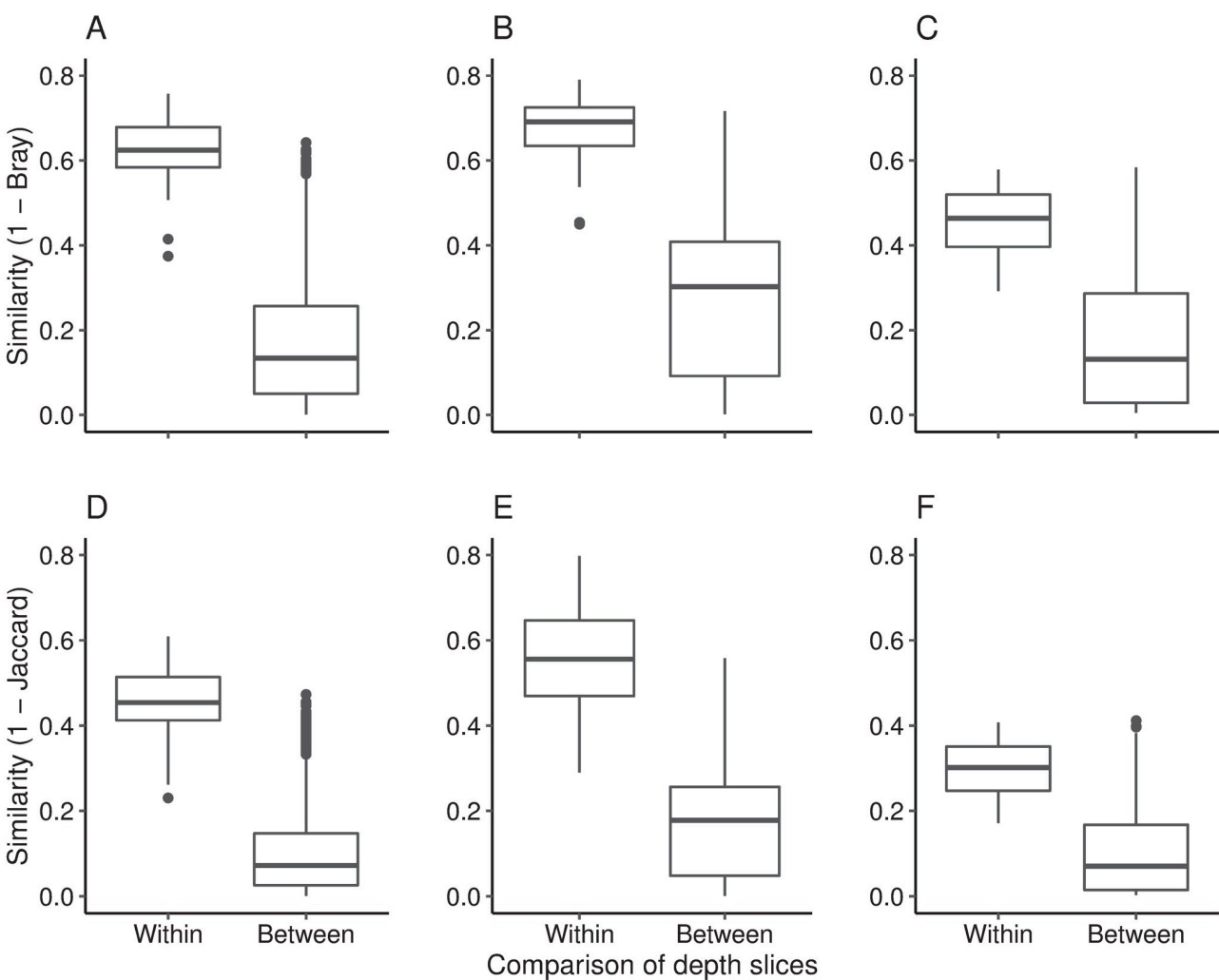

**Fig 2.** Similarity comparisons (1—Bray Curtis distance) within and between depth slices for Lake Nganoke (A), Lake Paringa (B), and Lake Pounui (C). Similarity comparisons (1—Jaccard distance) within and between depth slices for Lake Nganoke (D), Lake Paringa (E), and Lake Pounui (F). Boxes display the first and third quartile spread of the data, with the line in the box indicating the median, the whiskers denoting the minimum and maximum values and the dots as outliers of the data.

core where the laminae are not horizontal and have a dip of 20–30 degrees (S2 Fig). Replicates taken in this part of the core may be sampling across the stratigraphy which would conse- quently lower the similarity and may explain the lack of clustering. In Lake Nganoke, one of the replicates from slice 6 was very similar in composition to the slice above. Reworking of sed- iment by living organisms has been documented and could account for this observation although there was 5 cm between these layers [22].

Sediments from a range of depths/ages was analysed to determine whether there was greater variability (lower similarity between triplicates) in older sediments. We hypothesised that greater variability would occur in older sediments and that this might be caused by stochastic differences in DNA degradation. However, this hypothesis was not supported in lakes Nga- noke and Paringa, although in Lake Paringa there was a significant positive relationship between similarity and slice depth (p = 0.0241, $r^2$ = 0.357; Fig 4) and a significant increase in the proportion of shared reads with depth was observed (p < 0.001, $r^2$ = 0.658; S8E Fig). This could indicate that DNA degradation in the core is removing the rarer low abundance ASVs,

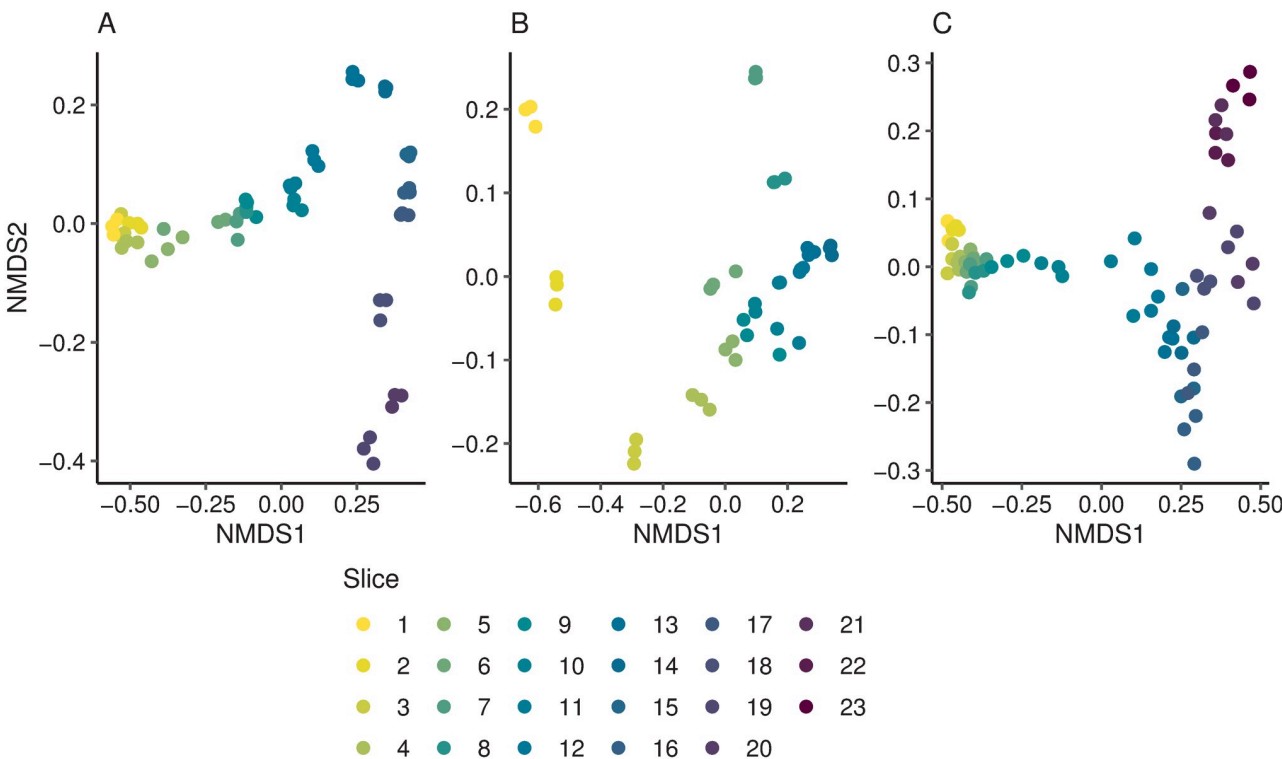

**Fig 3. Non-metric multidimensional scaling plot based on Bray Curtis distance matrices for samples taken from Lake Nganoke (A), Lake Paringa (B), and Lake Pounui (C).** Points are coloured by depth slice. Approximate ages of slices vary between lakes and are given in S1 Table. Stress values for the non-metric multidimensional scaling (nMDS) plots were Nganoke: 0.048; Paringa: 0.066 and Pounui: 0.055.

and that these deeper slices are thus dominated by abundant ASVs that are shared amongst all replicates. It is unknown why this significant trend is only observed in Lake Paringa and further investigation are required including assessing the effect of how different sediment matrices affect DNA degradation [20, 21].

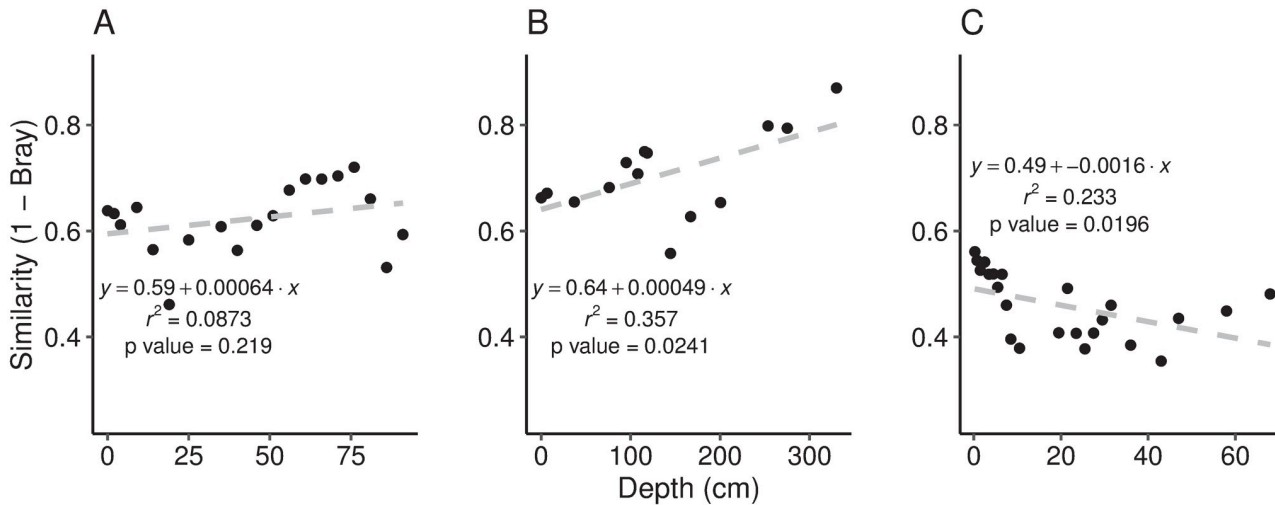

**Fig 4. Linear regression of similarity (1-Bray Curtis) and depth slice in Lake Nganoke (A), Lake Paringa (B), and Lake Pounui (C).** Depth slices are organised by depth (1 being the most recent). Approximate ages of depth slices vary between lakes and are given in S1 Table.

Only Lake Pounui showed a significant decline in similarity within a slice with depth (p = 0.0196, $r^2$ = 0.233; Fig 4). The decrease in similarity at these depths is most likely due to the non-horizontal laminae deeper in the core (S2 Fig), which would likely result in the replicates representing different time periods. If laminae are not horizontal then increased variability in replication may be observed and results would need to be carefully considered as replicates may represent different temporal periods.

## Conclusion

In summary, replication improved the detection of rare species in the sediment samples, however the full diversity was not captured even when triplicate samples were used in the surface depth slices. However, most paleolimnology studies do not aim to identify rare species, but rather explore broad scale shifts in community structure [6, 28]. There may be exceptions to this when investigating rare or invasive species [66], and in these instances it might be more appropriate to use targeted methods such as droplet digital PCR or refrain from subsampling if using metabarcoding [67].

The community structure and composition were more similar within depth slices than between them. However, there were exceptions where one replicate was markedly different. Reasons for this are unknown but could be related to bioturbation or sediment heterogeneity. This highlights the need for caution when interpreting these types of data and further replication and increased sampling of depths immediately above and below may be required. Variability among replicates remained relatively constant among depth and did not decrease as anticipated. The exception was Lake Pounui were there was a decrease in similarity with depth which was most likely related to laminae not being horizontal at these depths.

## Supporting information

**S1 Fig. Map of the position of each lakes.** The map was produced in ggmap [32] using Map tiles by Stamen Design, under CC BY 3.0. Data by OpenStreetMap, under ODbL.
(TIF)

**S2 Fig. Photographs of the sediment cores sampled with the sampled depth slices depicted.**
(TIF)

**S3 Fig. Accumulation curves displaying the observed number of Amplicon Sequence Variants (ASVs) with increasing number of reads for each distinct depth slice (see S1 Table for depths of slices) in sediment core samples taken from Lake Nganoke.**
(TIF)

**S4 Fig. Accumulation curves displaying the observed number of Amplicon Sequence Variants (ASVs) with increasing number of reads for each distinct depth slice (see S1 Table for depths of slices) in sediment core samples taken from Lake Paringa.**
(TIF)

**S5 Fig. Accumulation curves displaying the observed number of Amplicon Sequence Variants (ASVs) with increasing number of reads for each distinct depth slice (see S1 Table for depths of slices) in sediment core samples taken from Lake Pounui.**
(TIF)

**S6 Fig. The total richness per slice (triplicates combined) with depth down the core.** A—Lake Nganoke; B—Lake Paringa; C- Lake Pounui.
(TIF)

**S7 Fig. Relative proportional composition of the community at phyla level with depth down the core.** A—Lake Nganoke; B—Lake Paringa; C- Lake Pounui. Phyla that did not account for on average > 1% of the community were not show so bars do not add up to 100%.
(TIF)

**S8 Fig. The proportion of shared Amplicon Sequence Variants (ASVs; A—Lake Nganoke; B—Lake Paringa; C—Lake Pounui) and reads (D—Lake Nganoke; E—Lake Paringa; F—Lake Pounui) amongst replicates within a depth slice plotted against depth.** Note the different scales.
(TIF)

**S1 Table. Sub-bottom depth and age of each depth slice for lakes Nganoke, Paringa and Pounui.**
(XLSX)

**S2 Table. Number of reads before and after removing contaminating sequences from controls by subtraction.**
(XLSX)

## Acknowledgments

The authors thank Laura Biessy (Cawthron Institute), Lizette Reyes (GNS Science) and Jake Parrish (Victoria University) for field and laboratory assistance. Ngāti Kahungunu ki Wairarapa (lakes Nganoke and Pounui) and Te Runanga o Makaawhio (Lake Paringa) are acknowledged for their support of this project. The landowners at lakes Nganoke and Pounui are thanked for access to the launching sites. The Department of Conservation is acknowledged for assistance with permitting (Lake Paringa).

## Author Contributions

**Conceptualization:** Jamie D. Howarth, Marcus J. Vandergoes, Susanna A. Wood.

**Data curation:** John K. Pearman.

**Formal analysis:** John K. Pearman.

**Funding acquisition:** Marcus J. Vandergoes, Susanna A. Wood.

**Investigation:** Jamie D. Howarth, Susanna A. Wood.

**Methodology:** Georgia Thomson-Laing, Lucy Thompson, Andrew Rees.

**Project administration:** Marcus J. Vandergoes, Susanna A. Wood.

**Visualization:** John K. Pearman.

**Writing – original draft:** John K. Pearman, Susanna A. Wood.

**Writing – review & editing:** John K. Pearman, Georgia Thomson-Laing, Jamie D. Howarth, Marcus J. Vandergoes, Lucy Thompson, Andrew Rees, Susanna A. Wood.

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
