## [Decision Letter · Decision Letter 0]

5 Feb 2021

PONE-D-20-32628

Investigating variability in microbial community composition in replicate environmental DNA samples down lake sediment cores

PLOS ONE

Dear Dr. Pearman,

Thank you for submitting your manuscript to PLOS ONE. After careful consideration, we feel that it has merit but does not fully meet PLOS ONE’s publication criteria as it currently stands. Therefore, we invite you to submit a revised version of the manuscript that addresses the points raised during the review process.

 Please see below.

We look forward to receiving your revised manuscript.

Kind regards,

Kamlesh Jangid, Ph.D

Academic Editor

PLOS ONE

Additional Editor Comments:

Dear Dr. Pearman,

Thank you for revising your manuscript. Given the extent of revision I invited a new expert to have an unbiased opinion in addition to an earlier one. As you'll notice, both reviewers are now positive about the work and its quality. However, there are concerns about the writing, especially in the introduction which gives a different impression (Rev1) and a few instances of interpretation of the results and how you consider the shared ASV's (Rev2). In addition, given the degree of typographical errors, I strongly suggest that you carefully proof-read before submission; it is disappointing. If you're ready to incorporate these changes, I would be pleased to consider this further. I look forward to receiving a revised version soon.

Best,

Kamlesh

Journal Requirements:

We note that one or more of the authors are employed by a commercial company: GNS Science.

2.1. Please provide an amended Funding Statement declaring this commercial affiliation, as well as a statement regarding the Role of Funders in your study. If the funding organization did not play a role in the study design, data collection and analysis, decision to publish, or preparation of the manuscript and only provided financial support in the form of authors' salaries and/or research materials, please review your statements relating to the author contributions, and ensure you have specifically and accurately indicated the role(s) that these authors had in your study. You can update author roles in the Author Contributions section of the online submission form.

2.2. Please also provide an updated Competing Interests Statement declaring this commercial affiliation along with any other relevant declarations relating to employment, consultancy, patents, products in development, or marketed products, etc.  

3. We note that Figure S1 in your submission contain map/satellite images which may be copyrighted. All PLOS content is published under the Creative Commons Attribution License (CC BY 4.0), which means that the manuscript, images, and Supporting Information files will be freely available online, and any third party is permitted to access, download, copy, distribute, and use these materials in any way, even commercially, with proper attribution. For these reasons, we cannot publish previously copyrighted maps or satellite images created using proprietary data, such as Google software (Google Maps, Street View, and Earth). For more information, see our copyright guidelines: http://journals.plos.org/plosone/s/licenses-and-copyright.

3.1.    You may seek permission from the original copyright holder of Figure S1 to publish the content specifically under the CC BY 4.0 license. 

3.2.    If you are unable to obtain permission from the original copyright holder to publish these figures under the CC BY 4.0 license or if the copyright holder’s requirements are incompatible with the CC BY 4.0 license, please either i) remove the figure or ii) supply a replacement figure that complies with the CC BY 4.0 license. Please check copyright information on all replacement figures and update the figure caption with source information. If applicable, please specify in the figure caption text when a figure is similar but not identical to the original image and is therefore for illustrative purposes only.

Reviewers' comments:

Reviewer's Responses to Questions

**Comments to the Author**

1. Is the manuscript technically sound, and do the data support the conclusions?

Reviewer #1: Yes

Reviewer #2: Yes

2. Has the statistical analysis been performed appropriately and rigorously? 

Reviewer #1: Yes

Reviewer #2: Yes

3. Have the authors made all data underlying the findings in their manuscript fully available?

Reviewer #1: Yes

Reviewer #2: Yes

4. Is the manuscript presented in an intelligible fashion and written in standard English?

Reviewer #1: Yes

Reviewer #2: Yes

5. Review Comments to the Author

Reviewer #1: Broad comment

The study is about the small-scale variability (spatial and depth) of lake sediment core microbial communities using ASVs from 16S rRNA amplicon sequencing. The Introduction section, however, seems to be more focused on the variability derived from DNA extraction and other sequencing procedures from lake sediment samples. Perhaps authors want to prepare the Introduction for better clarity including a clearly stated study objective and more straight-forward hypotheses.

There are several occasions different terms used for the same concept, and results are presented differently between the main text and figure legend (see below). These are signs of the under-development of the manuscript. Relying on a review system for this kind of issues can’t be justified as the journal review system is already stretched very thin.

Specific comments

L24-25: Yes this is what the study was about, but the Introduction section seemed to more focus on the issue of DNA extraction variability and lack of study from lake sediment samples, so a more explicit statement to clearly show the overall objective is appreciated.

L76-78: Perhaps the authors want to address these by referring to the previously cited relevant studies in that they are mostly ignored the method-based variability. If the latter part of the statement means that lake sediments samples haven’t been assessed the method-based variability, the authors may also want to rewrite the latter half as the current version simply indicate that lake sediments samples DNA extractions haven’t been replicated, which is hard to believe.

L89-91: Hypothesis 1 is actually two hypotheses and the wordings are confusing, please consider rewriting it for more clarity.

L137-138: Were Lake Pounui samples stored at 4C until processed?

Sample collection: How many total sediment core samples were collected and processed per lake?

L234, 240: Authors used rarefaction curve in other places e.g., Methods section and Fig. 1 legend, although the species-accumulation curve is a better term to describe instead in general (or ASV-accumulation curve). Be consistent.

L237-238: Although this is correct, the hyperdiversity is not specific to ASVs as most microbiome studies using OTU also show similar rarefaction curve patterns.

L245: Change “could be” to “is”. In fact, a saturated rarefaction curve is a prime indication of low diversity.

L256-258: I would expect the opposite from the disturbances like climate change and anthropogenic impacts. Cite some references to support the authors’ claim here.

Figure S8. Add “among replicates” after “The proportion of shared ASVs” and “reads”.

L270-273: The Figure S8 legend says figures D-F are the shared reads among replicates, but here the statements are confusing. Consider rewriting for clarity and consistency.

L288-289: Both Bray-Curtis and Jaccard coefficients are widely used dissimilarity coefficients and correctly indicated in the Methods section. Collaborate how 1-J would indicate community structure.

L291-297: This section is more relevant to Figures 3 and S7. Rearrange accordingly.

L298-311: To me, the linear trajectories of samples according to the depth profile is more interesting than comparing among-replicates variability in NMDS space, which can be stochastic or artifact from sequencing technology or NMDS procedure itself. Consider identifying patterns according to the depth profiles within each lake and among lakes, and discuss potential determining factors.

L320-324: The positive relationship between similarity and depth seems to be even stronger if the depth is truncated at shallow (less than 100 cm) to be similar to two other lakes, so this argument is irrelevant.

L325-329: This is different from the original reasoning for the hypothesis. Do authors have any evidence to speculate this?

L338: Yes for the increasing sequencing depth, but rarefaction may still be very relevant as it can control possible sequencing depth variability among samples.

Conclusion: I would appreciate a more concise conclusion section corresponding to the overall objective and listed hypotheses, instead of summarizing the Results and Discussion section.

Reviewer #2: The authors have satisfactorily addressed most of my comments during their revision. Experimentally, this is really nice work, and small changes in the presentation would make it much more impactful. It also still needs a careful proofreading to avoid errors.

While a number of suggestions are on the attached manuscript, including some minor typographical errors, some of the major points are below.

Line 269. This is really not intuitive because ASVs are obtained from the reads, so the number of shared ASVs should not be very different from the number of shared reads. The difference is that when calculating the number of shared ASVs, you are not weighing them according to the number of reads. This heavily weighs them towards the rare reads, which in my opinion is misleading. It would be better to just report shared ASVs but weigh them according to the number of reads. This would be much more meaningful because the ASVs corrects potential sequencing errors. It would also reduce the number of figures, which would nice for the readers. If you don't want to do this, at least explain what you are doing and why the two measures are so different.

Figure 3 is brutal, and it is nearly impossible to connect the depth with the colors of the dots in the plots. Please recolor the dots with a continuous spectrum so that samples from similar depths are similar in color. This could be done either by varying the intensity or the color according to the spectrum (such as blue-green-yellow).

The paper would have been improved by running some technical replicates, ie. same sediment processed in parallel to determine the differences. Note that since the rarefaction curves did not overlap for most of the replicates, the populations were obviously different. It would be nice to if this was an experimental artifact or real. For instance, in soil, replicates are often different, but just because the soil communities are different even from similar samples. To late for these experiments, but maybe next time.

6. PLOS authors have the option to publish the peer review history of their article (what does this mean?). If published, this will include your full peer review and any attached files.

Reviewer #1: No

Reviewer #2: No

---

## [Author Response · Author response to Decision Letter 0]

23 Feb 2021

Reviewer #1: Broad comment

The study is about the small-scale variability (spatial and depth) of lake sediment core microbial communities using ASVs from 16S rRNA amplicon sequencing. The Introduction section, however, seems to be more focused on the variability derived from DNA extraction and other sequencing procedures from lake sediment samples. Perhaps authors want to prepare the Introduction for better clarity including a clearly stated study objective and more straight-forward hypotheses.

There are several occasions different terms used for the same concept, and results are presented differently between the main text and figure legend (see below). These are signs of the under-development of the manuscript. Relying on a review system for this kind of issues can’t be justified as the journal review system is already stretched very thin.

Authors: We thank the reviewer for their comments which have markedly improved the manuscript. We have now slightly refocused the introduction as suggested (see below for more details). We apologise for the inconsistencies in terminology, this has now been corrected throughout.

Specific comments

L24-25: Yes this is what the study was about, but the Introduction section seemed to more focus on the issue of DNA extraction variability and lack of study from lake sediment samples, so a more explicit statement to clearly show the overall objective is appreciated.

Authors: We have added in a sentence at the end of the introduction to further highlight the overall objective of the study

L76-78: Perhaps the authors want to address these by referring to the previously cited relevant studies in that they are mostly ignored the method-based variability. If the latter part of the statement means that lake sediments samples haven’t been assessed the method-based variability, the authors may also want to rewrite the latter half as the current version simply indicate that lake sediments samples DNA extractions haven’t been replicated, which is hard to believe.

Authors: We have reworked the order of the introduction slightly to aid the flow. We have also deleted the sentences of concern from the manuscript as these were not accurately portraying what we were aimed to highlight as noted by the reviewer

L89-91: Hypothesis 1 is actually two hypotheses and the wordings are confusing, please consider rewriting it for more clarity.

Authors: We agree, and have split the hypothesis in two. 

L137-138: Were Lake Pounui samples stored at 4C until processed?

Authors: These samples were processed within 6 hours and then stored at -80C. We have now added this information to the manuscript. L141

Sample collection: How many total sediment core samples were collected and processed per lake?

Authors: A single core per lake was collected. Three triplicate samples were taken at each depth slice in the core – this is stated in the methods. L142

L234, 240: Authors used rarefaction curve in other places e.g., Methods section and Fig. 1 legend, although the species-accumulation curve is a better term to describe instead in general (or ASV-accumulation curve). Be consistent.

Authors: We have changed to accumulation curves as suggested by the reviewer.

L237-238: Although this is correct, the hyperdiversity is not specific to ASVs as most microbiome studies using OTU also show similar rarefaction curve patterns.

Authors: We realise this is not specific to ASVs, although it is generally amplified compared to OTUs as there is no similarity clustering. We have changed the sentence to say molecular methodologies and added additional references. L242-243

L245: Change “could be” to “is”. In fact, a saturated rarefaction curve is a prime indication of low diversity.

Authors: Changed as requested.

L256-258: I would expect the opposite from the disturbances like climate change and anthropogenic impacts. Cite some references to support the authors’ claim here.

Authors: We have changed this sentence slightly and added in a reference suggesting that climate change has led to an increase in photosynthetic microbial taxa. L265-267

Figure S8. Add “among replicates” after “The proportion of shared ASVs” and “reads”.

Authors: Added as requested.

L270-273: The Figure S8 legend says figures D-F are the shared reads among replicates, but here the statements are confusing. Consider rewriting for clarity and consistency.

Authors: This section refer to the number of shared ASVs amongst replicates within a depth slice. We have clarified this in the manuscript and legends by adding the phrase “amongst replicates within a depth slice”.

L288-289: Both Bray-Curtis and Jaccard coefficients are widely used dissimilarity coefficients and correctly indicated in the Methods section. Collaborate how 1-J would indicate community structure.

Authors: Here we were referring to the similarity in the presence/absence of ASVs by using 1- Jaccard which we termed community structure. To make it clearer in the manuscript we have now used the term presence/absence. 

L291-297: This section is more relevant to Figures 3 and S7. Rearrange accordingly.

Authors: Thank you for pointing out the better position for this section. It has been moved accordingly. 

L298-311: To me, the linear trajectories of samples according to the depth profile is more interesting than comparing among-replicates variability in NMDS space, which can be stochastic or artifact from sequencing technology or NMDS procedure itself. Consider identifying patterns according to the depth profiles within each lake and among lakes, and discuss potential determining factors.

Authors: We thank the reviewer for this comment and the authors are in agreement with the point that the changes across depth profiles are of interest. Indeed, they are currently under investigation on a lake by lake basis in concert with other proxies to explain changes in lake communities. This manuscript focuses on methodological issues as we believed these need to be addressed initially. 

L320-324: The positive relationship between similarity and depth seems to be even stronger if the depth is truncated at shallow (less than 100 cm) to be similar to two other lakes, so this argument is irrelevant.

Authors: We have removed the reference to depth/age. We also added in a sentence saying that further investigation is required as to why the positive trend was only seen in a single lake and that the composition of the sediment matrix should be investigated. 

L325-329: This is different from the original reasoning for the hypothesis. Do authors have any evidence to speculate this?

Authors: We based this conclusion on an assessment of the photo of the core (Figure S2). We have now noted this in the manuscript. L322

L338: Yes for the increasing sequencing depth, but rarefaction may still be very relevant as it can control possible sequencing depth variability among samples.

Authors: Our key point in this sentence was that if during analysis the main aim is to determine presence/absence of a species (such as in the detection of invasive species) assessing all the data (i.e., without subsampling) may be beneficial in detecting rare occurrences. 

Conclusion: I would appreciate a more concise conclusion section corresponding to the overall objective and listed hypotheses, instead of summarizing the Results and Discussion section.

Authors: Thank you for this comment. We have adapted the conclusions making the references to the hypothesis and ensuring that this section is more concise. 

Reviewer #2: The authors have satisfactorily addressed most of my comments during their revision. Experimentally, this is really nice work, and small changes in the presentation would make it much more impactful. It also still needs a careful proofreading to avoid errors.

Authors: We thank the reviewer for their comments and we have now undertaken a careful proofread

While a number of suggestions are on the attached manuscript, including some minor typographical errors, some of the major points are below.

Authors: Thank you for pointing these out and they have been addressed

Line 269. This is really not intuitive because ASVs are obtained from the reads, so the number of shared ASVs should not be very different from the number of shared reads. The difference is that when calculating the number of shared ASVs, you are not weighing them according to the number of reads. This heavily weighs them towards the rare reads, which in my opinion is misleading. It would be better to just report shared ASVs but weigh them according to the number of reads. This would be much more meaningful because the ASVs corrects potential sequencing errors. It would also reduce the number of figures, which would nice for the readers. If you don't want to do this, at least explain what you are doing and why the two measures are so different.

Authors: We have added in a sentence to explain that the abundant ASVs are detected amongst replicates but there are a large number of low abundance and rare ASVs in the samples which accounts for the low number of shared ASVs. L284-285

Figure 3 is brutal, and it is nearly impossible to connect the depth with the colors of the dots in the plots. Please recolor the dots with a continuous spectrum so that samples from similar depths are similar in color. This could be done either by varying the intensity or the color according to the spectrum (such as blue-green-yellow).

Authors: We agree the colour palette used in the figure was not the easiest to visualise. We have changed the figure as suggested by the reviewer.

The paper would have been improved by running some technical replicates, ie. same sediment processed in parallel to determine the differences. Note that since the rarefaction curves did not overlap for most of the replicates, the populations were obviously different. It would be nice to if this was an experimental artifact or real. For instance, in soil, replicates are often different, but just because the soil communities are different even from similar samples. To late for these experiments, but maybe next time.

Authors: We agree with the reviewer that technical replicates would have also give more information on the methodological aspects of sampling cores. We have now added in a sentence that this should be done in the future to get a better idea of richness estimates. L273-276

Figure 1: Explain why only two lines show for some.

Authors: There are three replicates for each depth slice. However, in a couple of the plots two lines overlap and thus it is hard to distinguish the third line. We have now noted this in the figure caption.

Table S1: Rename worksheet 2

Authors: Apologies worksheet two should not have been present.

Figure S3-5: Reword legends

Authors: We have now improved these.

---

## [Decision Letter · Decision Letter 1]

14 Apr 2021

Investigating variability in microbial community composition in replicate environmental DNA samples down lake sediment cores

PONE-D-20-32628R1

Dear Dr. Pearman,

We’re pleased to inform you that your manuscript has been judged scientifically suitable for publication and will be formally accepted for publication once it meets all outstanding technical requirements.

Kind regards,

Hideyuki Doi

Academic Editor

PLOS ONE

Additional Editor Comments (optional):

I and the previous reviewer carefully checked the revised manuscript as well as the response letter. We agree the revisions according to the reviewers’ comments. I now can recommend to publish the paper in this journal.

Reviewers' comments:

Reviewer's Responses to Questions

**Comments to the Author**

1. If the authors have adequately addressed your comments raised in a previous round of review and you feel that this manuscript is now acceptable for publication, you may indicate that here to bypass the “Comments to the Author” section, enter your conflict of interest statement in the “Confidential to Editor” section, and submit your "Accept" recommendation.

Reviewer #1: All comments have been addressed

2. Is the manuscript technically sound, and do the data support the conclusions?

Reviewer #1: Yes

3. Has the statistical analysis been performed appropriately and rigorously? 

Reviewer #1: Yes

4. Have the authors made all data underlying the findings in their manuscript fully available?

Reviewer #1: Yes

5. Is the manuscript presented in an intelligible fashion and written in standard English?

Reviewer #1: Yes

6. Review Comments to the Author

Reviewer #1: I appreciate the constructive communications with authors as they have adequately addressed all my comments.

7. PLOS authors have the option to publish the peer review history of their article (what does this mean?). If published, this will include your full peer review and any attached files.

Reviewer #1: No

---

## [Editor Report · Acceptance letter]

22 Apr 2021

PONE-D-20-32628R1 

Investigating variability in microbial community composition in replicate environmental DNA samples down lake sediment cores 

Dear Dr. Pearman:

I'm pleased to inform you that your manuscript has been deemed suitable for publication in PLOS ONE. Congratulations! Your manuscript is now with our production department. 

Kind regards, 

on behalf of

Dr. Hideyuki Doi 

Academic Editor

PLOS ONE